# Cyanidin-3-O-glucoside as a Nutrigenomic Factor in Type 2 Diabetes and Its Prominent Impact on Health

**DOI:** 10.3390/ijms24119765

**Published:** 2023-06-05

**Authors:** Iga Bartel, Magdalena Koszarska, Nina Strzałkowska, Nikolay T. Tzvetkov, Dongdong Wang, Jarosław O. Horbańczuk, Agnieszka Wierzbicka, Atanas G. Atanasov, Artur Jóźwik

**Affiliations:** 1Institute of Genetics and Animal Biotechnology, Polish Academy of Sciences, 05-552 Jastrzębiec, Poland; i.bartel@igbzpan.pl (I.B.); n.strzalkowska@igbzpan.pl (N.S.); j.horbanczuk@igbzpan.pl (J.O.H.); a.wierzbicka@igbzpan.pl (A.W.); atanas.atanasov@dhps.lbg.ac.at (A.G.A.); 2Department of Biochemical Pharmacology and Drug Design, Institute of Molecular Biology “Roumen Tsanev”, Bulgarian Academy of Sciences, Acad. G. Bonchev Str., Bl. 21, 1113 Sofia, Bulgaria; ntzvetkov@gmx.de; 3Centre for Metabolism, Obesity and Diabetes Research, Department of Medicine, McMaster University, Hamilton, ON L8S 4L8, Canada; wangd123@mcmaster.ca; 4Ludwig Boltzmann Institute Digital Health and Patient Safety, Medical University of Vienna, Spitalgasse 23, 1090 Vienna, Austria

**Keywords:** cyanidin-3-O-glucoside, anthocyanins, polyphenols, bioactive compounds, type 2 diabetes, nutrigenomics, civilization diseases

## Abstract

Type 2 diabetes (T2D) accounts for a global health problem. It is a complex disease as a result of the combination of environmental as well as genetic factors. Morbidity is still increasing across the world. One of the possibilities for the prevention and mitigation of the negative consequences of type 2 diabetes is a nutritional diet rich in bioactive compounds such as polyphenols. This review is focused on cyanidin-3-O-glucosidase (C3G), which belongs to the anthocyanins subclass, and its anti-diabetic properties. There are numerous pieces of evidence that C3G exerts positive effects on diabetic parameters, including in vitro and in vivo studies. It is involved in alleviating inflammation, reducing blood glucose, controlling postprandial hyperglycemia, and gene expression related to the development of T2D. C3G is one of the beneficial polyphenolic compounds that may help to overcome the public health problems associated with T2D.

## 1. Introduction

In recent years, the development of nutrigenomics’ research has markedly progressed. This scientific field provides knowledge for the better understanding of the nutritional impact on health status. Knowledge of how a particular food affects metabolic pathways and gene regulation processes has great potential in the prevention and mitigation of many diseases, including type 2 diabetes (T2D) [1].

Diabetes is one of the civilization diseases and is also a chronic disease [2]. The morbidity trend of diabetes is still growing and accounts for a grave and global health burden. The development of T2D takes place under the influence of a combination of environmental factors and genetic predispositions [3]. Although the hereditary aspects are significant in the development of diabetes, because the same environmental exposures do not affect everyone in the same pattern, lifestyle is also important. There are many modifiable risk factors, such as overweight and obesity, sedentary lifestyle, daily stress, etc., which may be crucial in the development of diabetes [4]. Diet is an especially important contributing factor. Providing high-quality food that is a source of antioxidant components may exert a positive impact on reducing the risk of developing diabetes and on improving the metabolic parameters in patients with T2D [5]. Among the intensively investigated bioactive compounds are anthocyanins which represent flavonoids, a group of polyphenols widely present in the plant kingdom. Anthocyanins are mainly found in berries where cyanidin-3-O-glucoside (C3G) is among the most abundant representatives. C3G demonstrates potential in combating oxidative stress, alleviating inflammation, and improving glucose metabolism in the context of diabetes. Its antioxidant and anti-inflammatory properties, along with its ability to modulate signaling pathways, contribute to its beneficial effects on glycemic control and insulin sensitivity [6]. The positive effect of this compound was demonstrated in numerous studies that suggested a protective role in various disorders, e.g., cardiovascular diseases (CVDs) [7], neurodegenerative diseases such as Alzheimer’s and Parkinson’s [8], gastrointestinal diseases [9], metabolic diseases such as obesity, dyslipidemia, hypertension, as well as hyperglycemia [10].

It has to be mentioned that there are already trials to develop antidiabetic medicine using synthetic iminosugars and sugar derivates acting as an antidiabetic agent [11]. The modified structures of carbohydrates present as potential mimetics of sugars that naturally occur in the food. They are significant for the progress of antiviral, antibacterial, and other activities. The most crucial classes of structurally modified carbohydrates are C-branched sugars, annulated sugars, as well as C-glycosides [12,13].

This review paper focuses on the role of cyanidin-3-O-glucoside (C3G) as an essential nutrigenomic factor in decreasing the risk of developing a civilization disease with the emphasis on activities counteracting type 2 diabetes (T2D).

## 2. Civilization Diseases

The number of people suffering from civilization diseases, also referred to as chronic diseases, is constantly increasing. This global trend has particularly been accelerating since the beginning of the 21st century when noncommunicable diseases (NCDs; chronic diseases that are not infectious agents and are not transmitted from person to person) became the main cause of death. According to the statistics presented by the World Health Organization (WHO) CVDs such as stroke and ischemic heart disease, cancers, chronic respiratory diseases, diabetes, osteoporosis, and obesity dominate among the civilization diseases. Alarming data show that 74% of all deaths in the world are caused by noncommunicable diseases [14]. On the top of the list are CVDs and they are responsible for 17.9 million deaths per year, while the places further down the list belong to cancers (9.3 million death cases per year), chronic respiratory diseases (4.1 million deaths per year), and diabetes (2 million deaths per year) [14]. The increasing global tendency of occurrence of civilization diseases primarily concerns developed and developing countries. Taking into account the reasons for the those diseases, a promoting role is played by a combination of genetic, physiological, socioeconomic, and environmental factors, as well as behavioral impacts. The consequences of urbanization and industrialization have significant meaning for public health due to changing patterns of diseases. Although urbanization has a beneficial economic effect, it is also a source of health issues. Insufficient accommodation, overpopulation, and high levels of air and water pollution are only some of the risk factors contributing to the global increase in NCDs [15]. Multiple studies exist which determine associations between urban living and noncommunicable diseases. The extensive empirical scrutiny by Goryakin et al. (2017) indicated evidence that urbanization is positively linked to the higher prevalence of diabetes [16]. The same positive relationships refer to cancers [17] and obesity, not only among adults [18,19,20] but also among children [21,22]. Furthermore, the negative impact of urbanization is observed in increasing morbidity from CVDs across the world. The evolution of urbanization has caused relevant changes in daily lifestyle including an imbalanced diet, lack of physical activity, and the spread of harmful habits such as smoking and abuse of alcohol. An unhealthy lifestyle has a negative influence on metabolic parameters that are risk factors for CVDs including raised blood pressure, overweight and obesity, high blood glucose level (hyperglycemia), and excess levels of fat in the blood (hyperlipidemia) [23].

## 3. Diabetes

Type 2 diabetes (T2D) became a global problem after a significant increase of 70% in the numbers of death since 2000 [23]. It is the most prevalent type of diabetes, accounting for 90% of all diabetes. According to the International Diabetes Federation, 537 million people suffer from diabetes, which is in the region of 1 in 10 people. Approximately 541 million people have impaired glucose tolerance. In turn, 232 million people are not aware of having diabetes. It is estimated that in 2030 type 2 diabetes morbidity will grow to around 642 million cases [24]. T2D is described as a disturbance of carbohydrate, lipid, and protein metabolism as a consequence of abnormal insulin secretion, insulin resistance in muscles, liver, and adipocytes, or a combination of both [25]. To maintain proper β-cells function, whose role is producing insulin, the cellular integrity must be assured. Insulin is synthesized in pro-insulin form and during the maturation period, the conformational modification is carried out by proteins in the endoplasmic reticulum (ER) [26]. Subsequently, pro-insulin is transferred to the Golgi apparatus (GA) to immature secretory vesicles and is split into peptide C and insulin [27]. After maturation, insulin is stored in granules until its release, caused by a high level of glucose [28]. The role of insulin is in stimulating glucose uptake, mainly in skeletal muscles and adipose tissue. Impaired glucose uptake by peripheral tissues results in a deteriorated rate of glucose metabolism. Insulin is also responsible for the storage stimulation of energy provided with diet as glycogen in hepatocytes and skeletal muscles after food intake. Moreover, insulin stimulates liver cells to produce and store triglycerides in adipose tissue. Insulin dysfunctions may lead to the elevation of free fatty acids (FFAs) in plasma and hinders their further metabolization to energy. The result of the mitochondrial oxidation of fatty acids, Acetyl-CoA, can be metabolized to ketone bodies in hepatocytes. The excess of FFAs and ketone bodies decreases glucose uptake by tissues, deteriorating the hyperglycemic status. When it comes to protein metabolism, this hormone increases the rate of protein synthesis and diminishes protein degradation. Therefore, the insufficient production of insulin may lead to the intensification of protein catabolism. The latter results in a raised level of amino acids in plasma [29]. The pathophysiology of T2D is linked to disturbances in the feedback loop between insulin functioning and insulin releasing causing inappropriately high blood glucose levels. Another reason is associated with β-cells dysfunction; however, in this case, insulin secretion is diminished limiting the organism’s ability to sustain physiological glucose levels. In turn, insulin resistance (IR) is one reason for increasing glucose production in the liver and reducing glucose uptake in organs such as muscle, liver, and adipose tissue. When both processes occur at the beginning of the pathogenesis and take part in the progress of T2D, β-cells disorder is often more severe than IR. Nevertheless, when β-cells’ dysfunction and IR occur, the excess glucose level is enhanced affecting the development of T2D [30,31].

### 3.1. Lifestyle Factors and T2D

T2D is a multifactorial disorder occurring due to genetic and environmental factors. The pathophysiological changes are observed in pancreatic β-cell dysfunction as well as increasing chronic inflammation that leads to the gradually hindered control of blood glucose levels [32]. Oxidative stress and incorrect daily dietary behavior play a pivotal role in the development of T2D [33]. The Western diet is highly caloric due to an excess amount of fats and carbohydrates that raise blood glucose and circulation of very-low-density lipoproteins (VLDLs), and chylomicrons (CMs), which are rich in triglycerides (TG). It is a cause of the increasing production of ROS and further defective prompting of inflammatory molecules. A long pro-oxidant status is involved in dysfunctions in mitochondria, endoplasmic reticulum (ER), and increase in NADPH oxidase (NOX), and also superoxidase (O_2_-) production [34,35]. The most prevalent dietary mistakes are linked to snacking sweets between meals; a shortage of intake of dietary fiber, unsaturated fatty acids, and legumes; and an excess intake of meat and meat products [36]. An especially excessive intake of simple carbohydrates which have a high-glycemic index, such as sugar and refined cereals, has a highly negative impact promoting the development of T2D. The consumption of fat also has a serious impact, mainly as a highly processed industrial food that is a source of trans-fatty acids and an excessive intake of omega-6 in comparison to omega-3 fatty acids [37]. A diet rich in omega-6 polyunsaturated fatty acids (PUFAs) favors inflammation and the development of numerous civilization diseases [38]. Epidemiologic studies have indicated a positive correlation between the raised plasma level of inflammatory cytokines, such as tumor necrosis factor α (TNF-α) and interleukin-6 (IL-6), with a higher risk of diabetes [39]. At the same time, an insufficient consumption of fruits and vegetables is observed that are sources of vitamins, minerals, and bioactivity compounds such as polyphenols [40]. An imbalance between antioxidants and reactive oxygen species (ROS) can lead to damage or apoptosis of pancreatic β-cells and deteriorate the secretion of insulin [41]. ROS are able to activate cellular signaling pathways such as specific protein kinase C (PKC) isoforms, including PKC-β_II_ or nuclear factor-κB (NF-κB), and afterward disturb insulin signaling pathways that cause insulin resistance [42]. An important factor is also physical activity. Systematic training increases the level of anti-inflammatory cytokines (e.g., IL-1 Receptor antagonist (IL-1Ra, which is antagonist of pro-inflammatory IL-1)) and a diminished level of pro-inflammatory cytokines such as IL-6 and IL-18 [43]. Other behavioral and lifestyle risk factors are insufficient—the smoking of cigarettes independent of other factors [44], older age, body mass index (BMI) of ≥25 kg per m², and a sedentary lifestyle [45].

### 3.2. Genetic and Epigenetic Factors and T2D

There is evidence that genetic predisposition is also a risk factor for type 2 diabetes. Children are at a higher risk of developing T2D if their mother rather than their father suffers from this disease [46]. Genome-wide association studies (GWASs) indicated that a single nucleotide polymorphism (SNP) in the genes *TCF7L2*, *FTO*, *CDKAL1*, *IGF2BP2*, *CDKN2B*, and more than 100 common variants of the respective SNPs are positively correlated with type 2 diabetes. However, only a few variants are found in exons that affect gene function, for example, in *SLC30A8* which is responsible for encoding the zinc transporter that is necessary to store insulin [47]. The majority of the identified variants are located in introns (more precisely in genetic loci) and they may affect the expression of a nearby gene, but this has only been confirmed for a few genes [48]. Many differences between diabetes genes among the populations are observed due to the location and frequency of risk alleles. *TCF7L2* gene variants, which are positively correlated with diabetes risk, are found in 20–30% of the Caucasian population, while only in 3–5% of Asians [49,50]. In turn, the variant in the *KCNQ1* gene, which is linked to diabetes in Asians, is not common among Caucasians [51]. The majority of found diabetes loci in Caucasians and Asians are associated with impaired pancreatic β-cell function; meanwhile, only a few are found which are related to insulin resistance or fasting insulin levels. Therefore, pancreatic β-cell dysfunction may be the main cause of the development of diabetes [44]. Although genetic predisposition is important, it should be mentioned that these polymorphisms increase the risk of developing type 2 diabetes by approximately 10–20%. Currently, a few possibilities are being considered to explain the heritability of diabetes, such as heterogeneity of the disease, and interactions between gene–environment as well as epigenetic mechanisms [52]. A recent study has shown that a combination of three gene variants such as *SLIT3*, *PLEKHA5*, and *PPP2R2C* is a risk factor for insulin resistance and increases the probability of becoming insulin resistant by 50%. Moreover, high coffee and caffeine intake (>10 cups/day or 220 mg caffeine/day) notably increases the risk of insulin resistance in subjects with these genetic variants [53]. In turn, other authors reported that several SNPs, such as rs79105258 and rs10252701 variants, are also associated with the risk of type 2 diabetes. However, in this case, a moderate intake of coffee (≥2 cups) significantly reduces the morbidity of this disease [54,55]. The development of epigenetics may help to understand the molecular association between genetics, environmental factors, and type 2 diabetes [56]. This scientific field explains at least part of the type 2 diabetes pathogenesis through the investigation of histone modifications, DNA methylation, and microRNAs influencing changes in gene function or the heritability alternation without changes in the nucleotide sequence, which can be passed from one cell generation to the next. The study on gestational diabetes mellitus (GDM) mice showed that the pups demonstrated hypermethylation and epigenetic downregulation of *IGF2* and *H19* genes involved in insulin sensitivity [57]. There is a necessity to conduct more studies to determine the role of epigenetics in the development of type 2 diabetes.

## 4. Polyphenols and Cyanidin-3-O-glucoside

Several of type 2 diabetes risk factors are modifiable and the majority of civilization diseases could be likely diminished by a rational diet providing pivotal nutritional ingredients [58]. One group of beneficial dietary natural compounds for maintaining the proper functioning of the human body in the treatment and prevention of this civilization disease are polyphenols [59,60]. Polyphenols are a widespread group of bioactive compounds that are secondary metabolites of plants. They are responsible for protection against harmful external factors, such as ultraviolet radiation, oxidants, and pathogens. The presence of polyphenols is identified in fruits, vegetables, legumes, whole grain products, green and black tea, coffee, red wine, and cocoa. Currently, the number of polyphenols is estimated at about 10,000 plant compounds due to their versatile structures including the aromatic ring system and diverse amount of phenol units [61,62]. Polyphenols are divided into several categories (Figure 1).

Flavonoids are most prevalent in the daily human diet. The subclasses of this group of compounds include flavonols, flavanones, flavanols, anthocyanins, isoflavones, and chalcones. Among the anthocyanins, six major forms are distinguished such as cyanidin, malvidin, delphinidin, petunidin, peonidin, and pelargonidin (Figure 2) that are found in many fruits, e.g., wild blackberry, chokeberry, blueberry, strawberry apple, orange, and vegetables such as red cabbage, asparagus, carrot, and cauliflower [65,66,67]. Cyanidin-3-O-glucoside (C3G; Figure 2) is among the most widespread anthocyanin in the plant kingdom.

The main sources of this compound are blue, red, and purple fruits and vegetables [68]. According to the reports in the literature, the content of cyanidin-3-O-glucoside (C3G) is as follows: strawberries (3.7 mg/100 g); blueberries range from 3.01 to 3.93 mg/100 g; chokeberries (1.7 mg/100 g); cranberries (0.7 mg/100 g); lingonberries (1.4 mg/100 g) [56]. C3G, in the same way as the rest of anthocyanins, is water soluble. The bioavailability of C3G was established to be approximately 12%, estimated from the recovery of a stable isotopically labeled anthocyanin tracer in the urine and breath of participants [69]. The first stage of the metabolism of C3G takes place in the oral cavity (Figure 3).

Many factors may initiate C3G salivary or enzymatic hydrolysis. Next, the C3G is partly absorbed by active diffusion by the gastric epithelial cells and along with transporters such as bilitranslocase transporter (BTL), sodium-dependent glucose transporter 1 (SGLT1), glucose transporter 1 (GLUT1) and glucose transporter 3 (GLUT3), and mono-carboxylated transporter 1 (MCT1) is transported to the small bowel. Exposure to the intestinal conditions leads to a decrease in C3G’s bioavailability by 40–50% [70]. After deglycosylation, cleavage of the heterocyclic flavylium ring, dihydroxylation, and decarboxylation, C3G is converted to quinoidal form and absorbed by intestinal epithelial cells. Afterwards, together with transporters such as SGLT1 and glucose transporter2 (GLUT2), cyanidin-3-O-glucoside and its derived metabolites are transformed in the large intestine by gut microbiota and absorbed by colonocytes. After spreading to the body tissues, the final step is excretion in urine and feces [71]. According to a recommendation by the Joint FAO/WHO Expert Committee on Food Additives (JECFA), a daily intake of 2.5 mg/kg from grape skins is safe [72]. Moreover, a clinical trial has demonstrated that providing a diet of even 320 mg/kg/day of anthocyanins, including C3G, is tolerable by organisms and beneficial against inflammatory responses [73]. Additionally, a meta-analysis by Gao et al. [74] that included five prospective cohort reports showed that a 7.5 mg daily dietary intake of anthocyanins would decrease the risk of developing type 2 diabetes by 5%.

### Role of Cyanidin-3-O-glucoside (C3G) in Diabetes

C3G exhibits versatile bioeffects counteracting diabetes (Figure 4). C3G has an impact on delaying the absorption of disaccharides by acting as an inhibitor of α-glucosidase, intestinal β-fructosidase (sucrase), and pancreatic α-amylase. Moreover, wild blackberry extract exerted inhibitory effects on β-glucosidases in vitro. It seems to be helpful in controlling postprandial hyperglycemia in diabetic patients [53,75,76]. Additionally, the 5 week administration of wild blackberry extract led to a reduction in glucose levels from 360 to 270 mg/dL (*p* < 0.05) in diabetic rats [77]. An experiment conducted by Sun et al. (2012) showed that a C3G-rich bayberry fruit extract had a positive effect on diabetic parameters in both in vivo and in vitro testing. It prevented β-cells’ death, improved their viability, and diminished ROS production in the mitochondrion. Moreover, it enhanced insulin-like growth factor II gene transcript levels and insulin protein in rat insulinoma cell line INS-1 due to the regulation of pancreatic duodenal homeobox 1 gene expression. In turn, decreased blood glucose was observed in a diabetic animal model after the administration of C3G [78]. The protective effect on secretory function was also confirmed in the rat insulinoma INS1E cell line and murine islets of Langerhans treated with palmitic acid. The study performed revealed that the expression of proteins such as BAX, which is known as a pro-apoptotic factor, and apoptotic markers such as cleaved caspase-3 are significantly decreased; meanwhile, the anti-apoptotic protein BCL2 is upregulated by C3G [79].

Recent research reports demonstrated a significant role of the rough endoplasmic reticulum (RER) in the development of diabetes. The RER present in the cytoplasm of pancreatic β-cells is an initial site of proinsulin molecule synthesis and is responsible for quality control of this process, including folding and disulfide pairing [80]. Many factors can disturb the function of RER and lead to the gathering of abnormal proteins. This state is known as ER stress. The pivotal transmembrane protein induced in response to stress is PERK (protein kinase RNA-like ER kinase), which is associated with β-cell death [81]. The C3G has the ability to inhibit the PERK pathway [73]. Anthocyanins, including C3G, can ameliorate diabetic parameters due to the activation of AMP-dependent protein kinase (AMPK). AMPK is a pivotal cellular energy sensor and is involved in controlling lipid metabolism, glucose homeostasis, and insulin sensitivity [82]. Excess blood glucose can inhibit AMPK phosphorylation and activity, disturbing its downstream signaling and leading to glycolysis and lipolysis that deteriorate diabetic disorders [83]. Incubation of cultured 3T3-L1 adipocytes under hyperglycemia in C3G presence enhances AMPK activity and reduces the amount of free fatty acids (FFAs) and glycerol [84]. It has been reported that C3G could upregulate the gene expression of PPARs (Peroxisome-proliferator activated receptors) [85] involved in the regulation of biological processes, such as lipoprotein metabolism [86], energy homeostasis, and glucose metabolism [87] depending on isoforms. PPARs are a common target involved in the beneficial outcome of many diverse dietary compounds [88]. For example, the activation of isoform ɣ leads to the enhancement of insulin sensitivity due to protection from the effect of TNF-α in adipocytes and improves the gene expression involved in glucose and lipid metabolism [89]. To demonstrate that a major and direct target protein for the C3G is PPARs, Jia et al. (2020) conducted a study using PPARα-deficient mice as an experimental group [73]. Mice were fed a high-fat diet and supplemented with C3G for 8 weeks. The results showed that parameters such as decreased plasma triglycerides and fasting glucose concentration in the control group were significantly lower; meanwhile, in the experimental group this effect was abrogated [90]. However, the molecular mechanism of action and direct target of C3G has to be further investigated. It has been shown recently that C3G is involved in restoring insulin signaling, the upregulation of *GLUT4* expression, and inhibition of mitochondrial stress. Developing insulin resistance and type 2 diabetes is linked to adipocyte disfunction [91]. A study on diabetic mice fed 0.2% of C3G as an addition to forage for 5 weeks revealed that mitigation of hyperglycemia and enhancement of insulin sensitivity was associated with reduction in retinol-binding protein 4 (RBP4) expression. Moreover, C3G upregulated the glucose transporter 4 (Glut4) in the white adipose tissue, which was related to the decrease in the inflammatory adipocytokines such as TNF-α, (IL-6) and monocyte chemotactic protein-1 (MCP-1) [92]. Previous studies have demonstrated that expression of Glut4 is decreased in diabetic patients [93,94]. C3G also affects adipogenesis in 3T3-L1 cells. The preadipocytes differentiated into smaller adipocytes after treatment with C3G. Furthermore, in adipocytes tested with 20 μM and 100 μM pure C3G the production of TNF-α was diminished to 57% and 32%, respectively. In turn, adiponectin secretion was higher proportionately, over two-fold and four-fold, respectively [95]. Adiponectin is a protein hormone and plays a pivotal role in energy metabolism due to the activation of glucose transport, inhibition of gluconeogenesis through AMPK, and mitigation of inflammation via the PPARα pathway [96]. In the same study, researchers demonstrated that 20 μM and 100 μM pure C3G significantly improved CCAAT/enhancer binding protein α (*C/EBPα*) gene expression to 150% and 341%, respectively, and glucose transporter 4 (*GLUT4*) gene expression over two-fold. This action was linked with ameliorated glucose uptake of 1.3- and 1.8-fold in adipocyte cell culture. C/EBPα and PPARα are transcription factors involved in adipocyte differentiation, which are downregulated by ALK7 leading to fat accumulation in metabolic diseases such as obesity [97].

## 5. Conclusions

C3G, a major dietary anthocyanin, shows positive effects on metabolic parameters in diabetes. A diet enriched in polyphenol food sources counteracts type 2 diabetes and possibly other civilization diseases. Application of C3G may be one of the promising therapeutic approaches to improve public health and overcome global problems linked to civilization diseases such as type 2 diabetes in the future.

## Figures and Tables

**Figure 1 ijms-24-09765-f001:**
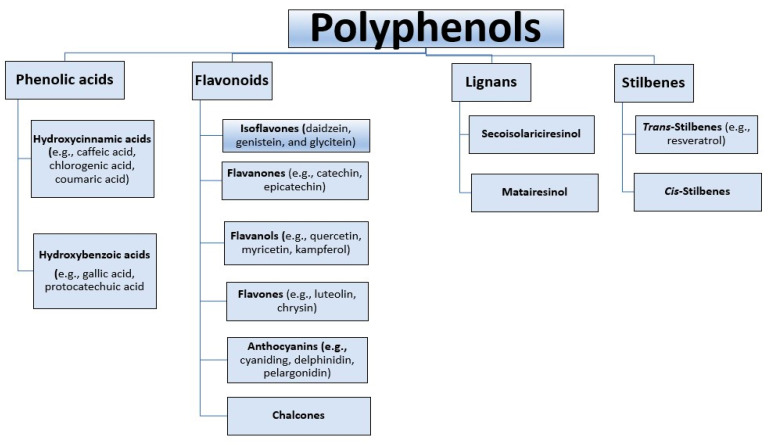
Classification of polyphenols [63,64].

**Figure 2 ijms-24-09765-f002:**
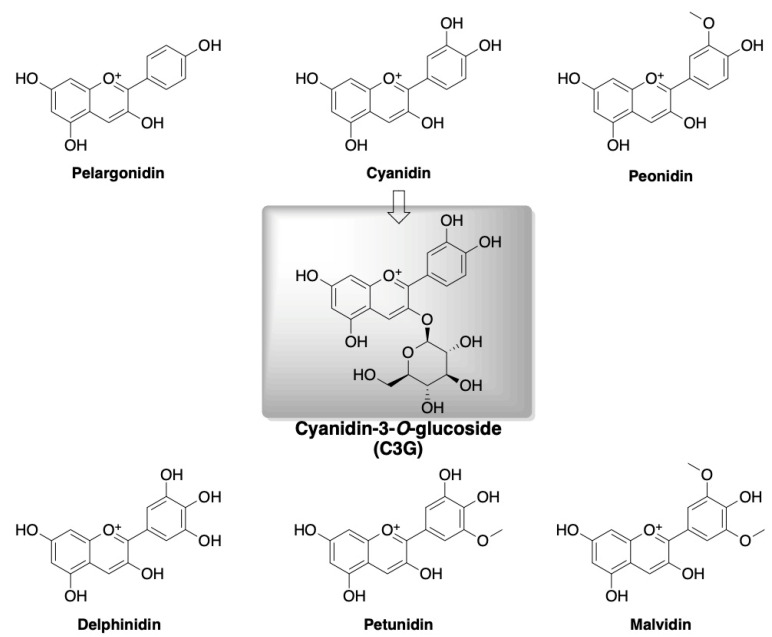
Chemical structures of major anthocyanins.

**Figure 3 ijms-24-09765-f003:**
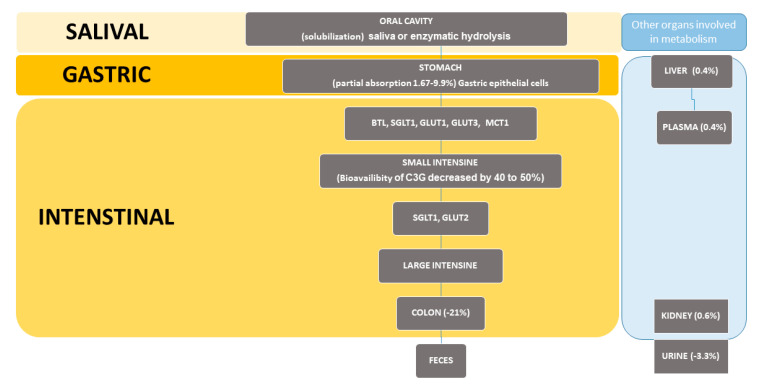
Scheme of the mechanism of metabolism and bioavailability of C3G.

**Figure 4 ijms-24-09765-f004:**
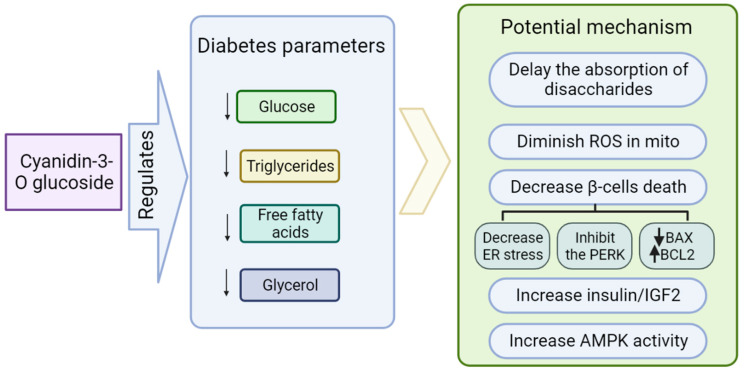
Bioactivities of C3G on diabetes. C3G improves diabetes probably through deceasing glucose [78], triglycerides [79], free fatty acid [80], and glycerol levels [81]. The further molecular mechanism of these effects may be related to the absorption of disaccharides, ROS in the mitochondria [82,83], β-cell death [84], insulin/IGF2 [85], and AMPK activity [86].

## Data Availability

Data sharing not applicable. No new data were created or analyzed in this study. Data sharing is not applicable to this article.

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
