# Peer review of "Cyanidin-3-O-glucoside as a Nutrigenomic Factor in Type 2 Diabetes and Its Prominent Impact on Health"

_ijms, 2023, doi:10.3390/ijms24119765_

Round 1
Reviewer 1 Report
Reviewer comments and suggestions
The authors in this review focused on cyanidin-3-O glucosidase (C3G), which belongs to the anthocyanins subclass, and exerts anti-diabetic properties. They have reported that with the help of various studies C3G exerts positive effects on diabetic parameters. It is involved in alleviating inflammation, lowering blood glucose, controlling postprandial hyperglycemia, and gene expression related to the development of T2D. Hence, the authors suggested that C3G have polyphenolic compounds, which may be beneficial for lowering public health problem associated with T2D.
Overall, the manuscript structure was weak, and needed a thorough revision in the manuscript. However, a few concerns/comments needed to be explained/modified.
- Line 69 Please mention the details of C3G.
- Line 84 The authors already used the term above, where they can use (CVD)
- Line 85-86 Needed a reference
- Line 92 is this important to highlight civilization words many times and in the title as well
- Line 110-11 Additionally the authors added homocysteine metabolism what was the purpose
- Comments for diabetes section: If the authors want to discuss the mechanism then it is better to include good papers in the manuscript and discuss.
- Comments for Figure 1 It’s a simple one, please add examples or elaborate more in the figures
- Section 4.1 It would be nice if they can divide the section based on glucose, lipid, and other diabetes parameters
- Comments for Figure 3 Discuss a few lines in the form of a legend and you can modify the figure by adding the authors et al with references.. the authors need to discuss more on the related topic.. Its show insufficient
- Please check the guidelines of mdpi, it seems that the authors need to modify all the references
Author Response
We thank you very much for the careful revision to our manuscript. We realized from the comments received that several key points in our original work still have not been properly addressed. In this resubmitted version, we modified the text accordingly, adding explanations and new citations.
Best regards,
Magdalena Koszarska

Reviewer 2 Report
In this review article, the authors summarized the role of cyanidin-3-O glucoside (C3G) as an essential nutrigenomic factor in decreasing the risk of developing civilization disease with emphasis on activities counteracting type 2 diabetes (T2D). My comments are as follows
1. The scope and importance of this review article is high due to the following reasons:
a) The review article emphasized on the prevention of T2DM using anthocyanin(C3G) derivatives which are from various readily available dietary sources which are known to act as antioxidants also.
b) Due to the usage of natural remedies, side effects will be very less as compared to the traditional medications.
c) The best part of C3G is that, it might prevent T2DM by means of different mechanistic pathways.
2. The review is well written and presented with rational scientific language with no typographical errors. The title and the abstract represent the objective of the manuscript.
3. The schemes and images are presented correctly (structures all molecules are correct) and texts presented are in line with them.
4. The section 4 describes about the C3G like sources, its metabolism, bioavailability etc. It will be better if presented in a schematic way (mechanism of metabolism)
5. In section 5 the authors describe the role of C3D reduction of T2DM reporting various in vivo as well as in vitro studies from various literatures, the reports are found to be authentic an could be convinced that C3G is a potential inhibitor.
The fig 3 showed that, C3G can prevent T2DM by means of various mechanistic pathways. So, the author may present literature reports against each mode of action in a tabular form with rational descriptions.
7. The references are not formatted in line with the journal standard, please correct them. Further, it is suggested to mention the importance of synthetic efforts towards development of antidiabetic medicine. In this regard, it is recommended to emphasis the importance of iminosugars and sugar derivatives as an antidiabetic agent and suggested to cite following relevant articles in the introduction section.
a. Po-Sen Tseng, Dr. Chennaiah Ande, Prof. Dr. Kelley W. Moremen, Prof. Dr. David Crich. Influence of Side Chain Conformation on the Activity of Glycosidase Inhibitors. Angewandte Chemie International Edition. 2022. (https://doi.org/10.1002/anie.202217809)
b. Rajasekaran, P.; Ande, C.; Vankar, Y. D. Synthesis of (5,6 & 6,6)-oxa-oxa annulated sugars as glycosidase inhibitors from 2-formyl galactal using iodocyclization as a key step. ARKIVOC 2022, vi, 5−23.
c. Chennaiah, A.; Bhowmick, S.; Vankar, Y. D. Conversion of glycals into vicinal-1,2-diazides and 1,2-(or 2,1)-azidoacetates using hypervalent iodine reagents and Me3SiN3. Application in the synthesis of N-glycopeptides, pseudo-trisaccharides and an iminosugar. RSC Adv. 2017, 7, 41755−41762.
Author Response
We thank you very much for the careful revision to our manuscript. We realized from the comments received that several key points in our original work still have not been properly addressed. In this resubmitted version, we modified the text accordingly, adding explanations and new citations.
Please see the attachment.
Best regards,
M.Koszarska
